# Hydrogen Therapy and Its Future Prospects for Ameliorating COVID-19: Clinical Applications, Efficacy, and Modality

**DOI:** 10.3390/biomedicines11071892

**Published:** 2023-07-04

**Authors:** Ishrat Perveen, Bakhtawar Bukhari, Mahwish Najeeb, Sumbal Nazir, Tallat Anwar Faridi, Muhammad Farooq, Qurat-ul-Ain Ahmad, Manal Abdel Haleem A. Abusalah, Thana’ Y. ALjaraedah, Wesal Yousef Alraei, Ali A. Rabaan, Kirnpal Kaur Banga Singh, Mai Abdel Haleem A. Abusalah

**Affiliations:** 1Food and Biotechnology Research Centre, Pakistan Council of Scientific and Industrial Research Centre, Lahore 54590, Pakistan; 2University Institute of Public Health, The University of Lahore, Lahore 54590, Pakistan; 3School of Zoology, Minhaj University Lahore, Lahore 54770, Pakistan; 4Division of Science and Technology, University of Education, Township Lahore, Lahore 54770, Pakistan; quratulainahmad@ue.edu.pk; 5Department of Medical Microbiology and Parasitology, School of Medical Sciences, Universiti Sains Malaysia, Kubang Kerian 16150, Malaysia; 6Department of Diet Therapy Technology & Dietetics, Faculty of Allied Medical Sciences, Zarqa University, Al-Zarqa 13132, Jordan; 7Molecular Diagnostic Laboratory, Johns Hopkins Aramco Healthcare, Dhahran 31311, Saudi Arabia; 8Department of Medical Laboratory Sciences, Faculty of Allied Medical Sciences, Zarqa University, Al-Zarqa 13132, Jordan

**Keywords:** molecular hydrogen, apoptosis, anti-inflammatory, reactive oxygen species, antioxidant, COVID-19

## Abstract

Molecular hydrogen is renowned as an odorless and colorless gas. The recommendations developed by China suggest that the inhalation of hydrogen molecules is currently advised in COVID-19 pneumonia treatment. The therapeutic effects of molecular hydrogens have been confirmed after numerous clinical trials and animal-model-based experiments, which have expounded that the low molecular weight of hydrogen enables it to easily diffuse and permeate through the cell membranes to produce a variety of biological impacts. A wide range of both chronic and acute inflammatory diseases, which may include sepsis, pancreatitis, respiratory disorders, autoimmune diseases, ischemia-reperfusion damages, etc. may be treated and prevented by using it. H_2_ can primarily be inoculated through inhalation, by drinking water (which already contains H_2_), or by administrating the injection of saline H_2_ in the body. It may play a pivotal role as an antioxidant, in regulating the immune system, in anti-inflammatory activities (mitochondrial energy metabolism), and cell death (apoptosis, pyroptosis, and autophagy) by reducing the formation of excessive reactive O_2_ species and modifying the transcription factors in the nuclei of the cells. However, the fundamental process of molecular hydrogen is still not entirely understood. Molecular hydrogen H_2_ has a promising future in therapeutics based on its safety and possible usefulness. The current review emphasizes the antioxidative, anti-apoptotic, and anti-inflammatory effects of hydrogen molecules along with the underlying principle and fundamental mechanism involved, with a prime focus on the coronavirus disease of 2019 (COVID-19). This review will also provide strategies and recommendations for the therapeutic and medicinal applications of the hydrogen molecule.

## 1. Introduction

The hydrogen (H_2_) molecule is widely known as the most prevalent and lightest element found in the atmosphere of the earth. It has, however, also been regarded as a novel naturally occurring antioxidant molecule having some propensity for interaction with many biomolecules and possible medicinal and therapeutic purposes. Hydreliox, known as the breathing gas, which is a mixture of H_2_, oxygen [1], and helium (He) gases, was widely recommended and administered as a source of H_2_ for therapeutic purposes, i.e., to avoid nitrogen narcosis and decompression sickness during extremely deep technical diving in humans. The first instance of H_2_ being utilized therapeutically took place in the late twentieth century when research on mice with cutaneous squamous carcinoma revealed that hyperbaric H_2_ significantly reduced tumor growth. The inhalation of the modest concentrations of H_2_ reduced the risk of ischemia-reperfusion (I/R), dramatic cerebral damage, and other cerebral strokes in mice by regulating oxidative stress [2].

Numerous cellular, animal, and clinical studies have found the biological effects of hydrogen (H_2_) molecules, focusing mostly on its anti-inflammatory, antiapoptotic, and antioxidative properties. It has been reported that the selective free radical and inflammatory scavenging ability of H_2_ is still widely acknowledged to be its mechanism, despite numerous errors. According to a prior study, the inhalation of H_2_ has been found to prevent acute pancreatitis in rats, which was reported to be induced by caerulein. This could happen by preventing oxidative stress and premature inflammation [3]. Patients suffering from various chronic pulmonary diseases have been reported to respond well to H_2_ treatment in clinical trials. This has confirmed that the therapeutic application of H_2_ is both safe and effective [4] Moreover, the therapeutic and medicinal applications of H_2_ have also been demonstrated in patients suffering from cardiac-arrest-related diseases due to oxidative stress and sports-related disorders [5,6]. Some of the therapeutic applications of hydrogen molecules have been given in Figure 1. It has been recommended to breathe H_2_ gas (66.6% H_2_) mixed with O_2_ (33.3% O_2_), since H_2_ plays a vital role in preventing lung function loss and emphysema and other lung conditions, according to a publication issued about the prevention of COVID-19 by the Health Commission of China of Clinical Guidance for Pneumonia Treatment [7].

H_2_ is portable, safe, and easy to supply, therefore drinking it may be better. H_2_ may dissolve in water at 1.6 mg/L (0.8 mM), without changing pH, at ambient temperature and atmospheric pressure. However, because H_2_ is poorly soluble in water, its low bioavailability may not give enough H_2_ in certain local injury conditions. H_2_ water injections help extend the half-life. H_2_-rich water retained 41% of H_2_ consumed by the body [8]. After drinking H_2_ water, typical H_2_ sensors may not detect enough H_2_ in rats’ brains [9]. H_2_ intervention works best when administered at high payloads to targeted areas. It was found that the microbubble (MB) delivery method places H_2_ gas on the MB shell and transports it through blood flow. H_2_ MBs had a higher H_2_ content/volume than H_2_-saturated saline, suggesting they may be better at preventing myocardial damage in mice [10]. Hydrogen baths have enhanced therapeutic implications because hydrogen may diffuse through the cell membrane. H_2_ water baths can heal skin disorders [11,12]. Hydrogen also preserves graft organs. Removed grafted organs were cold-preserved in H_2_-rich saturated water to minimize chronic graft-vs.-host disease and cold I/R graft injury [13,14]. 

Post-COVID-19 syndrome’s symptoms are characterized by ongoing, disappearing, recurring or relapsing symptoms, developed over more than twenty days, followed by the development of infection. It can, however, manifest as severe, moderate or mild symptoms [15,16,17]. Individuals are likely to be predisposed to post-COVID-19 syndrome due to a combination of factors, including an overactive immune system, chronic inflammation, tissue damage brought on by infection, and stress resulting from the pandemic’s concurrent socioeconomic effects [18,19]. An accurate diagnosis of this condition is challenging, however, because of the viral infection’s resolution and the absence of serological antibodies [16,20]. Chronic fatigue, which has been reported in more than fifty percent (58%) of cases, cognitive deficits, myalgia, and dyspnea are among the common symptomatic manifestations [21]. The currently available data indicate that symptoms that make it difficult for patients to carry out daily tasks can affect up to 63% of individuals in post-COVID-19, of which 17.8% of patients were working before developing COVID-19 [7]. It was found that nosocomial patients tend to perform routine tasks less efficiently than outpatients [22]. Despite the initial appearance of mild symptoms, post-COVID-19 syndrome symptoms develop in children. According to compiled research, 10% of children aged 2 to 11 have one or more COVID-19-related persistent symptoms, and this number rises to 13% for adolescents aged 12 to 16 years old [7]. COVID-19 may be a factor in the post-COVID-19 syndrome’s persistent immune dysregulation and major organ dysfunction, because of the systemic inflammation and hyperactive immunological responses. Chronic fatigue, cognitive deficits, and cardio-respiratory dysfunction are the most common symptoms of post-COVID-19 syndrome. More than fifty percent of the reported cases indicate chronic fatigue, exertional fatigue, and OxS as the most prevalent symptoms. Research suggests that this condition is also characterized by persistent mitochondrial dysfunction, OxS, and inflammation. The symptoms of myalgic encephalomyelitis (ME) and chronic fatigue (CFS) syndrome, which is a highly individualized disorder, can include cardiovascular distress (such as palpitations and irregular heartbeat), cognitive dysfunction (such as anxiety, confusion, decreased cognitive function, and forgetfulness), dizziness, and extreme fatigue. Since research on the long-term implications of H_2_ on COVID-19 and/or SARS-CoV-2 infections is still in its early stages, this review analyses the recent studies and proposes its positive future prospects. Furthermore, the current review emphasizes the anti-apoptotic, antioxidative, and anti-inflammatory properties of hydrogen (H_2_) molecules, as well as the underlying principle and fundamental mechanisms involved, with a prime focus on the coronavirus outbreak of 2019. This review will also provide strategies and recommendations for the therapeutic and medicinal applications of the hydrogen molecule. 

## 2. Biological Effects of Microbial Hydrogen

### 2.1. Role of Physiology of H_2_ Molecule in Therapeutic Applications

Natural molecular hydrogen is tiny, inert, and colorless. Airborne H_2_ gas burns at 4–75% concentrations. It quickly diffuses into the blood via alveoli during breathing and distributes throughout the body. H_2_ molecules penetrate through the cellular membrane quickly and distribute to organelles including the cytoplasm, nucleus, and others to perform biological tasks due to their molecular weight, and non-polar nature. H_2_ molecules can easily cross most of the barriers (i.e., blood–brain) which is not possible for most antioxidant molecules. H_2_ has no known cytotoxicity. Hydrogen molecules do not affect blood pressure, pH, or body temperature [23]. Mammalian cells, however, lack hydrogenases, which prevents them from producing molecular hydrogen. It has been reported that acute pulmonary injuries and distress syndromes may include respiratory distress and lung injury, i.e., ARDS and ALI, respectively. These may exhibit the characteristics of alveolar proteinaceous exudate, pulmonary edema, dysregulated inflammation-induced endothelium and epithelial damages, and alveolar/capillary barrier breakdown. The interstitium and bronchoalveolar area receive neutrophils, as given in Figure 2. The 2012 Berlin Conference classified ARDS as mild, moderate, or severe based on hypoxemia severity. Many lung insults can cause ALI. Dysregulated oxidative stress, apoptosis, and autophagy also cause ALI [24]. This is illustrated in Figure 2. 

### 2.2. Administration Routes and Exposure to Hydrogen Molecules

The administration of H_2_, and the delivery procedures that are typically used in animal models and human research studies, may include the inhaling of H_2_ gas, the consumption of hydrogen-dissolved H_2_O molecules, and the inoculation of H_2_ saline in the body. Systems for delivering nanomaterials have also been created recently. However, the effects of all distribution methods depend on how well H_2_ dissolves in liquids such as water, saline, or blood. The various administration routes for molecular hydrogen to combat various body infections and/or diseases, along with their respective management strategies, are given in Table 1. H_2_ gas inhalation is the simplest therapeutic and has been utilized extensively since the first report. H_2_ inhalation guarantees the dose and retention time in the body. H_2_ that has been inhaled can travel throughout the body via the circulatory system and may diffuse into the plasma through the alveoli. Clinical testing revealed that 72 h of exposure to 2.4% H_2_ gas had no negative impacts on any physiological measures, which indicates that the H_2_ molecule does not have any negative effects on the human body [25]. The chemical components and features of the H_2_ molecule, however, indicate that it reacts with oxygen to generate water when it burns. According to research, H_2_ does not explode when mixed with air or oxygen if the concentration is less than 10%, even if it may be explosive and deadly when it is higher than 4% in the air [26]. Additionally, the research has demonstrated that the H_2_ concentration in both tissues and blood depends on the intensity and time of inhalation. Moreover, the antioxidant action of H_2_ was also found to be dose-dependent [27]. Recently, it has become more and more popular to provide a gaseous mixture (i.e., H_2_:O_2_; 66:33%) produced through the electrolysis of H_2_O, which has found practical implications in both clinical and research evaluations [28]. High concentrations of H_2_ gases may be used to provide more beneficial effects. High-pressure gas cylinders can be safely and conveniently replaced with a generator that does not need to be restocked [29].

When retinal damage (I/R) showed a transient rise in intraocular pressure and reactive oxygen species (ROS), the prolonged administration of H_2_-saturated eye drops prevented apoptosis in animal models [39]. The saline injection of molecular hydrogen is a well-known method that directly applies H_2_ to the affected area and quickly delivers a large amount of H_2_. H_2_ injections can be harmful. H_2_ was delivered orally, intravenously, intraperitoneally, or inhaled in mice. Gas chromatography, with high-quality sensors, measured H_2_ in various tissues. Thus, molecular H_2_ can independently reach most human organs or blood via these three methods [9].

In pigs, hydrogen gas inhalation and its pharmacokinetics showed that the peak of molecular H_2_ saturation was lower in venous blood than arterial blood, indicating the diffusion of H_2_ molecules during bloodstream transport [40]. Mitochondrial respiration, xanthine oxidoreductase, and NADH/NADPH oxidase produce ROS, such as hydroxyl (•OH), superoxide anion (O_2_•), peroxyl (RO_2_•), nitric oxide (NO•), and alkoxyl radicals [41]. Cell injury hinders electron transport and mitochondrial oxidative phosphorylation, leaking electrons to produce excess ROS. ROS damage cellular or organelle membranes. Figure 3 shows how lipid peroxidation after membrane release produces leukotrienes and arachidonic acid, which tend to produce inflammation. Neutrophils and macrophages may produce ROS to destroy infections, damaging healthy cells’ mitochondria and nuclei and killing them [42]. 

For many pathogenic processes, hydrogen peaks in the oral and inhalation administration steps, but the duration of drinking H_2_ water was longer [43]. These ROS comprise hydroxyl (•OH), superoxide anion (O_2_•), peroxyl (RO_2_•), nitric oxide (NO•), and alkoxyl (RO•) radicals, and are generally formed by NADH/NADPH oxidase, mitochondrial respiration, and/or xanthine oxidoreductase [41]. Electron transport and mitochondrial oxidative phosphorylation are hampered by cell damage, and electrons leak out to produce an excessive amount of ROS. On the one hand, cell or organelle membranes are harmed by excessive ROS production. Leukotrienes and arachidonic acid, which have been reported to support inflammatory pain, are created by the lipids’ subsequent peroxidation after they have been released from the membrane. Furthermore, neutrophils and macrophages may create ROS to kill infections, which may damage healthy cells’ mitochondria and nuclei and ultimately lead to death [42].

## 3. Biological Effects of Hydrogen

### 3.1. Antioxidant Effect

#### 3.1.1. ROS Neutralization

Oxidative stress is the common first step in many processes implicated in many illnesses and it is reportedly caused by a disparity between the antioxidant system and ROS [43]. These ROS comprise hydroxyl (•OH), alkoxyl (RO•), nitric oxide (NO•), superoxide anion (O_2_•^−^), and peroxyl (RO_2_•) radicals. It has been found that they are typically produced by NADH/NADPH oxidase, xanthine oxidoreductase, and/or mitochondrial respiration. When cells are damaged, electron transport and oxidative phosphorylation in the mitochondria are hampered, and electrons leak to form excessive ROS. This excessive ROS generation damages cellular or organelle membranes. The lipids are then separated from the membrane and peroxidized, producing arachidonic acid and leukotrienes, both of which contribute to ameliorating inflammatory pain. Additionally, ROS which have been generated by macrophages and neutrophils may tend to attack pathogens. This would further cause severe damages to the cellular organelles, including nuclei and mitochondria, and may subsequently initiate cellular apoptosis. H_2_ as a reductant can permeate and neutralize the cellular membrane against harmful substances and particles which may be found in the cellular structure (•OH and ONOO) and essentially negates the impacts of O_2_ and H_2_O_2_ in maintaining the internal environment stability and various physiological functions. A proposed method of action was the scavenging of (•OH) radical by the chemical fusion of the hydrogen molecule with the hydroxyl ion and ultimately producing a water molecule and hydrogen ion, which was later followed by the fusion of the hydrogen ion with the oxygen molecule leading to the production of HO_2_ [44]. 

Molecular hydrogen can protect against I/R damage by lowering the scavenging of ONOO and OH and oxidative stress, which function as ROS’ electron donor molecules, but only in acellular tests. After two weeks of breathing 1.3% H_2_ gas, vasculitis mice had less OH and ONOO, reducing tissue damage. It also prevents hydroxyl radicals from undergoing the Haber–Weiss and Fenton reaction to create •OH radicals [45]. 

The antioxidant potential and biological benefits of H_2_ persist after elimination, especially at lower levels [46]. This suggests that the process involves regulating antioxidant signals rather than scavenging free radicals. H_2_-rich saline administration stimulates the Nrf2-ARE signaling pathway, reducing experimental autoimmune encephalomyelitis (EAE) symptoms in mice [47]. 

The *Alternaria alternata* tangerine pathotype illustrates ROS detoxification signaling mechanisms. ROS resistance genes are activated by H_2_O_2_ from the membrane-bound NADPH oxidase (NOX) complex. When exposed to ROS, YAP1 conformationally changes, forms disulfide bonds with two conserved cysteine residues, and enters the nucleus to regulate environmental stress genes. ROS detoxification requires the YAP1 and HOG1 MAP kinase, SKN7 redox-responsive regulators, NOX complex, Siderophores, and NPS6-mediated siderophore synthesis, which absorbs iron from the environment and requires NPS6’s non-ribosomal peptide synthetase [48]. This is illustrated in Figure 4. 

In addition, intracellular ROS is significantly reduced by the activation of Nrf_2_ transcription which increases the SOD glutathione synthesis and downregulates the expression of NADPH oxidase [49]. Hydrogen may prevent cell death by preventing aberrant phospholipid oxidation, and lipid peroxidation, as well as by limiting the rise in cell membrane permeability, which is yet another crucial mechanism of H_2_ antioxidation [50]. Interestingly, significant recent studies have shown that high antioxidant levels increased the mortality rates from cardiovascular disease and cancer. An ideal antioxidant should reduce oxidative stress without disrupting redox equilibrium [51]. Due to its fast diffusion into cells through blood circulation, H_2_ may serve as the optimal antioxidant [52,53].

#### 3.1.2. Regulation of Mitochondria

Along with the methods by which H_2_ neutralizes oxidative stress, the mechanisms leading up to the malfunction of the electron transport chain—the first step of mitochondrial oxidative stress—were emphasized. As they generate 90% of a cell’s energy in the form of ATP, mitochondria are sometimes referred to as the powerhouses of the cell. The production of ROS via forward and reverse electron transfer is accompanied by this mechanism, which depends on oxidative phosphorylation [54]. By limiting excessive hydrogen production, H_2_ reduces mitochondrial dysfunction. It is believed that the leakage of electrons from the electron chain transport may be able to repair the cells’ malfunctioning.

The mitochondria are where the ATP-sensitive K+ channel (mKATP), a crucial player in energy control, is present. To balance the amount of cardiac NAD+ and the generation of ATP (mitochondrial), which would lessen myocardial I/R damage, H_2_ gas might activate mKATP and control mitochondrial membrane potential [55]. One of the most important elements of the mitochondrial sequence of electron transfers is Coenzyme-Q [56]. In humans, CoQ10 predominates, but in rats, CoQ9 does. CoQ helps in the formation of NAD+, and proton motive force, both of which function as the ATP precursors by accepting electrons from Complex I and Complex II and transferring them to Complex III [57]. Nivolumab’s clinical effectiveness may be improved by H_2_ gas by boosting the amount of CoQ10 in mitochondria and replenishing worn-out CD8+ T cells’ action. Thus, it has been presumed that, through enhancing mitochondrial activity, H_2_ can prevent cell damage. The correction of mitochondrial dysfunction is anticipated to also enhance the disorganized signal transmission that influences the process of cellular death, for instance, in Caspase and Bax actions [58].

Mitophagy is essential for maintaining homeostasis in mitochondria [59]. The homeostasis is maintained by removing malfunctioning and damaged mitochondria. A mitophagy receptor named Fundc1 (protein 1), that controls mitophagy and interacts with LC3 II to support the maintenance of ATP balance in the mitochondria, is found on the cellular surface of mitochondria. The administration of H_2_ (2%) for three hours, was found to increase Fundc1-induced mitophagy. This resulted in rescuing the mice against liver damage, which was induced by sepsis. Moreover, H_2_ has a neuroprotective impact on glucose/oxygen-deprivation-induced brain injury in mice, which was found to increase in the expression of the mitophagy-associated genes, Parkin and PINK1 [60]. This further suggests that the advantageous role of hydrogen in ATP production could be due to the stimulation of mitochondrial autophagy. The mitochondrial malfunctioning, as reported by various sepsis-based animal studies, may diminish cellular energy which may further cause the failure of multiple organs. 

H_2_ treatment, for instance, upregulated heat shock protein 32 (HO-1; heme oxygenase-1) in cardiac tissues, scavenging ROS and preventing sepsis-related damage to multiple organs in a HO-1/Nrf_2_ dependent pathway [61]. Many neurodegenerative diseases are primarily brought about by excessive ROS-induced mitochondrial damage [62]. Previous research has demonstrated that H_2_ intervention has antioxidative effects on animals suffering from Parkinson’s and Alzheimer’s disease [63].

This is despite the fact that the breathing of 6.5% H_2_ gas at 2 L/min twice daily for an hour had no positive effects on those with Parkinson’s disease. It was postulated that this is related to H_2_ concentration and treatment duration and concluded by the speculation that H_2_ may balance mitochondrial electron transport, which would account for its ability to improve mitochondrial energy metabolism and scavenge ROS [64].

### 3.2. Anti-Inflammatory Effect

External pathogenic infection or tissue damage is considered to trigger inflammation, which is the body’s adaptive response [65]. Inflammation may lead to an increase in monocytes, neutrophil, as well as other immune cells, in addition to the generation of inflammatory cytokines. Mononuclear phagocytes and lymphocytes may travel from veins to the region of injured tissue, where they can develop and activate into macrophages. The primary source of cytokines and growth factors in this mechanism is phagocytes [21]. Inflammatory transcription factors such as nuclear, hypoxia-inducible, matrix metalloproteinases, nitrosyl radicals, and apoptotic factors (e.g., NF-B, HIF-1, and p53) may all be triggered by excessive intracellular ROS [66,67,68]. As a consequence, several reciprocal effects of cellular damage, inflammation, and apoptosis might coexist throughout the pathogenic phase of oxidative stress. By inhibiting the production of intercellular adhesion and chemokine molecules, hydrogen may prevent neutrophil and macrophage invasion during the initial stages of inflammation [69], for instance, by inhibiting the production of IL-1β and TNF-α (inflammatory cytokines), which tends to subsequently reduce inflammatory cytokines such as IFN-γ and IL-6 [70]. 

H_2_-rich serum l levels of IL-6, TNF, and IL-1 blocked the activation of the critical inflammatory signaling pathway NF-B, which, in turn, reduced the airway and pulmonary inflammatory response which is caused by any burn in mice [71]. Additionally, H_2_ has been shown to significantly lower NF-B expression in a variety of injury models, including acute sports injuries to skeletal muscle injury [22], liver injury, and hematencephalon [22]. This implies that H_2_ molecules can influence the inflammation process through the various regulating and modulating factors which are involved in nuclear transcription and proinflammatory cytokines. Additionally, it is important to highlight the balance between pro- and anti-inflammation while treating disorders caused by dysfunctional inflammation. The anti-inflammatory effects of H_2_ can also be observed in the animals suffering from cerebral injury (I/R) and allergic rhinitis by regulating Tregs (T-cells), which cause a reduction in the NF-B expression along with having an immunosuppressive effect [28,34].

Heme oxygenase-1 has been reported as a microsomal enzyme (rate limiting) and heat-shock protein that is involved in heme catabolism. Bilirubin, a powerful endogenous antioxidant, is produced when biliverdin is rapidly reduced. It may lower NF-B and IL-1 expression, hence reducing septic damage [72]. H_2_ infusion enhanced the synthesis of anti-inflammatory cytokine (i.e., IL-10) and HO-1 in mouse lung tissue and endothelial cells from human umbilical veins that had been activated by LPS [73]. It has also been demonstrated that pre-inhaling H_2_ gas can effectively prevent the onset of acute forms of pancreatitis by promoting the early expression and production of a heat stress protein (Hsp60) in mice, which promotes synthesis in response to high temperatures in order to defend and protect itself [3]. Due to this, it is thought that hydrogen may boost the body’s defenses and significantly aid in the anti-inflammatory process.

## 4. Hydrogen (H_2_) and Cell Death Regulation 

### 4.1. Apoptosis

Cell shrinkage, the formation of apoptotic bodies, and the condensation of chromatin are all characteristics of a type of planned cell death known as apoptosis. As a result, cells are cleared from the body while causing little injury to neighboring tissues, which is critical for tissue homeostasis and regulating cellular turnover [22]. Both internal and extrinsic cues can cause apoptosis. The cell surface’s death receptors activate the extrinsic apoptotic cascade by interacting with the Fas and tumor necrosis receptor factors, resulting in the caspase-8 being activated and, eventually, apoptosis. The antiapoptotic proteins, B-cell-lymphoma-2, and proapoptotic Bax were all found to be associated with the intrinsic apoptotic pathway [41].

Both apoptotic routes meet at a similar location, resulting in DNA fragmentation and caspase-3 activation [74]. H_2_ may have an antiapoptotic impact via scavenging ROS or regulating gene transcription, and both of these may influence endogenous apoptosis. In the in vitro investigation carried out in the epithelial cells of the intestine, it has reportedly been found that caspase-9 and 3 were suppressed but that cell viability was retained, and ROS production was dramatically reduced by H_2_-rich media. Moreover, H_2_ reversed the overexpression of Bcl-2 and Bax [75].

This impact of hydrogen-enriched water can be accomplished by preventing the mitochondrial translocation of apoptotic markers Bax and caspase-3. H_2_-rich water may potentially have an antiapoptotic effect by increasing the production of Bcl-2, a key factor (antiapoptotic), as shown in Figure 5. Furthermore, by stimulating the mitogen-activated protein kinase (MAPK)/HO-1 pathway, H_2_ can reduce ischemic brain damage in newborn mice and decrease neuronal death [76]. Alternatively, by stimulating the PI3K/Akt signaling pathway, alveolar epithelium (protect type II) cells protect against hyperoxia-induced apoptosis [77]. 

By stimulating the production of cleaved caspase-3, H_2_ has been demonstrated to increase cell death and inhibit the growth and migration of lung and esophageal cancer cells, which indicates a possible use of H_2_ in tumor therapy [78]. Hence, it has been proposed, through recent research, that H_2_ may serve numerous roles, including shielding normal cells from harm and limiting cancer cell growth.

### 4.2. Autophagy

By digesting macromolecules, autophagy can support, but it can also exacerbate, tissues and organ’s damage and inflammation, as seen in sepsis. Beclin-1 and LC3 protein are two autophagy-related proteins that play critical roles in autophagy detection. It has been demonstrated that H_2_ protected cardiomyocytes from isoproterenol-induced damage by suppressing autophagy [79]. In LPS-induced lung damage, H_2_-saturated water dramatically decreased the indication of LC3 and Beclin-1 (autophagy proteins) which indicates that tissues are sheltered by H_2_, preventing excessive autophagy [80]. H_2_ might, however, relieve LPS-induced neuroinflammation by lowering mTOR expression in glial cells, inducing autophagy and raising the ratio of LC3 II with LC3 I. This may be because of the varying intensity of the models occupying LPS-induced inflammation [22]. By adjusting mitophagy, mitochondrial ATP balance can be maintained with the help of a receptor such as Fundc-l. A three-hour treatment with 2% H_2_ protected mice against sepsis-induced liver damage and increased Fundc1-induced mitophagy [81]. This is shown in Figure 6.

Moreover, it has been demonstrated by investigations that there is an increasing trend in the Beclin-1 voicing of impaired cardiomyocytes and ratio of LC3 I/LC3II when H_2_-rich water was present, showing that H_2_ was involved in the breakdown of flawed mitochondria to maintain intracellular homeostasis [30]. By inhibiting the p38 and JNK/MAPK stress pathways, H_2_ can also promote autophagy [7]. Cell apoptosis and autophagy were also considerably increased in the cell lines of H1975 and A549 of lung cancer cured by using various doses of H_2_ gas [82].

Thus, it has been concluded that H_2_ followed a bidirectional regulating influence for autophagy hyperactivated during inflammation and/or can provide immense protection to tissues and cells from any harm.

### 4.3. Pyrolysis

Pyrolysis is a kind of controlled cell death which protects monocytes, microphages, and various pathogens. Caspase-1 is required for pyrolysis activation, and the primary downstream inflammatory agents in the pyrolytic pathway are cytokines, IL-1 and IL-18. H_2_ has been shown to have antibacterial properties in septic mice [83,84]. Saline enriched with H_2_ can substantially reduce the expression of caspase-1, which subsequently reduces the inflammatory response in early subarachnoid hemorrhage brain damage models [85]. Additionally, in models of organ damage due to sepsis, H_2_ therapy dramatically decreased caspase-1 expression in the injured organ as well as IL-18 and IL-1 cytokine levels. We already knew that H_2_ lung expansion is a good means of preventing I/R damage to donor lungs. However, H_2_ could have a specific regulatory function in malignancies [86]. 

Although there has been no direct evidence which may tend to explain the process of hydrogen involved in cell pyroptosis, it is likely that the H_2_ modulation of various nuclear and inflammatory components will interfere with pyroptosis progression. The actions of H_2_ on the pyrolysis route may suppress the production of tumor cells and/or provide protection for normal cells and tissues from harm, which is analogous to apoptosis.

## 5. Therapeutic Administration of H_2_ Molecules and Its Application in COVID-19

Pneumonia, pulmonary fibrosis, severe bronchitis, and COVID-19 have been widely known as severe pulmonary syndromes. COVID-19, however, has reportedly been found to rapidly spread all over the world. It is currently responsible for almost 185 million verified cases. The majority of COVID-19 patients show as having just a respiratory infection, beginning with a dry cough and fever and progressing to breathing difficulties and respiratory failures. Around 80% of ailed persons recover without hospitalization, whereas the other 20% suffer pneumonia, and approximately 5% develop acute ARDS [87]. Presently, only a few of medicines have been shown to promptly alleviate respiratory signs and limit disease development.

Mobilization in infiltrating immune cells and alveolar macrophages is increased with the rise in infections, causing prion inflammatory cytokines to be released into the alveoli and bronchioles. Alveolar hypoxia activates inflammatory pathways, resulting in the production of ROS and stimulation of hypoxia inducible and nuclear (NF-B) and (HIF-1) factors [88].

Mitochondrial ROS production typically initiates when cellular injury takes place, which can result in the destruction of the alveolar epithelial cell membrane and the surface deactivation, increasing membrane permeability and resulting in increased protein leakage in alveoli in the time of lung injury [89,90]. While high-speed oxygen breathing is possible, airway inflammation and exudation of viscous mucus in alveoli and bronchioles may render blood oxygenation in severe COVID-19 ineffective, because O_2_ cannot easily permeate in mucus plugs. Due to its low molecular weight, H_2_ has the potential to increase forced vital capacity while decreasing overall respiratory system resistance [91]. Moreover, H_2_ gas may improve dyspnea in COPD patients by decreasing bronchiole mucus buildup and hyperplasia in goblet cells [92].

Furthermore, in individuals with low SpO_2_ levels, the breathing of high concentrations of oxygen may cause damaging superoxide free radicals, which may lead to paralyzing lung function. As a result, for patients of COVID-19, inhaling H_2_ may be an effective way to combat both oxidative stress and hypoxia, lowering downstream cytokine release. Antioxidants such as Vitamin-E and co-enzyme Q-10, i.e., coenzyme Q-10, have been recommended to prevent lung surfactants from lipid peroxidation [93]. SARS-CoV-2 in the bronchus activates the immune system. Monocytes and lymphocytes enter the alveoli through small capillaries and release excessive cytokines, i.e., IL-6 and TNF-, triggering cytokine storms and damaging the alveolar epithelial cells. However, when NF-B transcription is inhibited, it reduces the activation of immune cells which may subsequently reduce hydrogen-induced inflammation. Hydrogen protects the epithelial cells of alveoli from apoptosis and oxidative stress by regulating Nrf2 transcription. Oxygen delivery and bronchial mucus production can also reduce dyspnea by using hydrogen [94]. This is shown in Figure 7. 

H_2_ therapeutics have gained popularity in the last two decades of objective analysis due to their simple and diverse application methods. Recent studies demonstrate the therapeutic, i.e., anti-inflammatory and antioxidant, characterization of molecular hydrogen. H_2_ therapeutics, such as oxy-hydrogen inhalation, can improve post-COVID parameters such as mild cognitive impairment, chronic fatigue, and cardiovascular function inhibition [87]. However, if H_2_ therapies are recommended as alternative or ancillary COVID-19 treatments, a comprehensive strategy including clinical evidence, cost-benefit analysis, dosage concentrations and durations, and further mechanistic studies will be needed. It has been found that the unique characteristics of molecular H_2_ i.e., its lessened molecular weight, electrochemical neutrality, and gaseous and non-polar nature, prevents electrochemical gradients, hydrophilic, and hydrophobic forces from affecting H_2_ distribution across phospholipid membranes [89,90]. H_2_ has a significant impact on processes occurring in the specific cellular structures, including organelles such as mitochondria [95]. H_2_-inclusive interventions, for a wide range of are both infectious and non-infectious purposes, are being studied in labs and clinics. H_2_ can be administered by inhalation, infusion, ingestion, or topical application [96]. 

Severe COVID-19 infections require anti-inflammatory and antioxidant therapy. An overloaded immune system can cause catastrophic inflammatory cytokine storms that damage the lungs. In COVID-19 patients, serum levels of IL-6 and IL-10 are also highly linked with disease severity, which indicates that inflammatory cytokines could be potential biomarkers. In an animal model, inhaling 2% H_2_ greatly decreased the number of cells which cause inflammation and TNF, IL-23, IL-6, and IL-17 gene levels in the broncho alveolar lavage fluid [97]. It has also been found that 45 min of H_2_ gas inhalation reduced airway and pulmonary inflammation in chronic obstructive pulmonary disease (COPD) and asthma patients by reducing MCP-1, IL-6, and IL-4 levels. Hence, it has been reported that the administration of H_2_ in COVID-19 patients might positively decrease cytokine hailstorms and, as a result, may reduce acute lung damage [4]. 

Different H_2_ dosages have been observed to reduce the oxidative stress biomarker MDA and raise the levels of antioxidant enzymes such as GSH in the blood and lung tissues of animal models of airway inflammation [86]. It has also been shown that H_2_-rich media intervention reduces damage in human cell lines (A549) of lung epithelial cells (irradiation-induced) by lowering ROS generation [98]. H_2_ reduces cell damage, ROS production, alveolar epithelial barrier degradation, and gas exchange across the alveoli [99]. As a result, we have grounds to think that, by neutralizing oxidative stress, H_2_ can effectively mitigate COVID-19 pneumonia. We noted that in a few recently reported multicenter clinical studies, the researchers employed a combination of H_2_ and oxygen [1] gas (66% H_2_; 33% O_2_), produced by electrolyzed water and supplied to patients with COVID-19. Even though randomization was not used because of the importance of dealing with the outbreak, a considerably larger proportion of patients in the therapy group were reported to inhale a mixture of H_2_ and O_2_, and showed improved clinical symptoms, than control group patients, who received traditional oxygen treatment [38].

Similarly, using H_2_ increases the O_2_ utilization rate, decreases O_2_ intake, as well as the negative impacts on exercise in healthy adults [97]. Inhaling an O_2_ and H_2_ mixture can enlarge the bronchioles and minimize inspiratory work, promoting O_2_ absorption through alveoli [23]. SARS-CoV-2 may cause lymphopenia by stimulating the p53 apoptotic signaling pathway in lymphocytes. H_2_ may prevent apoptosis in peripheral blood cells which could potentially assist in COVID-19 [33]. To reduce lung damage, surfactant proteins can be enhanced by H_2_ [100]. In combination with the preceding studies, we propose that H_2_ inhalation may be a potential treatment for COVID-19 by decreasing inflammation, apoptosis, hypoxia, and oxidative stress to some extent.

## 6. Effects on Human Immune System

The overactivation of immune system cells and prion inflammatory chemicals plays a significant role in the development of inflammation in many inflammatory disorders. The traditional animal model for multiple sclerosis in humans is EAE. H_2_-rich water intervention may improve EAE symptoms by reducing CD4+T cell infiltration and suppressing Th17 cell growth in the spinal cord [35]. Different H_2_ concentrations for immune paucity can enhance the immune deficiency condition and antitumor immunological activity by increasing the percentage of CD^8+^ T-cells [101].

The most frequent side effect in many people receiving radiation is immunological dysfunction. According to studies, pretreatment with H_2_-enhanced CD^8+^ and CD^4+^ T cells prevented radiation-induced splenocyte death in mice, which prevented immunological dysfunction [102]. In addition, healthy adult peripheral blood cells showed a considerable downregulation of inflammation and apoptotic signaling following four weeks of H_2_-water intake. When eosinophils and mast cells are activated, type I hypersensitivity reactions, that result in allergic rhinitis, produce tissue congestion and edema. Around a 67% concentration of hydrogen molecules can alleviate it by preventing Th_2_ cells from responding in an inflammatory response [103]. Moreover, it has been found that H_2_-rich saline reduces allergic rhinitis by rectifying the Th_1_/Th_1_ polarity. Many inflammatory disorders have pathogenic characteristics of macrophage circulation and M_1_/M_2_ disproportion [104].

Acute kidney damage [105], ischemic stroke [106], and rheumatoid arthritis [107] are only a few conditions for which high concentrations of H_2_ have been shown to significantly enhance IL-4 via controlling the M_1_/M_2_ balance. In a chronic pancreatitis rat model, H_2_ was first shown to reverse Treg loss, demonstrating that H_2_ also controls inflammation through mediating Treg [34]. By encouraging Treg proliferation and preventing immunological overactivation, a low dosage H_2_ intervention decreased inflammation [108]. As a result, the various H_2_ dosages may regulate the proliferation of immune cells to balance immunological overactivation or immunodeficiency. 

Recent studies have shown that H_2_ inhalation has a negative impact on the process number, duration, and immunohistochemical signals of microglia in rats with chronic L-DOPA-induced striatal lesions. Reactive microglia produce a variety of cytokines and chemokines inflammatory compounds, as well as cell surface molecules, that are primarily responsible for macrophagic and antigen-cell function [34,108]. Some findings explicitly demonstrate that inhaling H_2_ was unable to bring astrocyte levels down to normal levels. It is critical to note that astrogliosis’ effects on adjacent non-neural and neural cells may be both positive and negative [34,108]. Findings from some other studies, however, suggest that regarding the impact of hydrogen on inflammatory conditions in conjunction with its impact on the microglial (striatal) reactivity may generally support the reduction in LID [34,108]. However, these findings could not prevent scientists from perpetually speculating about the other mechanisms involved in the significant anti-dyskinetic effect of molecular hydrogen.

The current study details how the numerous effects of H_2_ on the treatment and prevention of various diseases is still in the initial phases. Clinical trials and efficacy evaluation on animals and cell cultures differ significantly, so further investigation is required. Evidently, it has been observed that every research investigation has yielded inconsistent results. However, the administration of hydrogen in the human body, and the various subsequent associated factors, such as excessive accumulation, reduction potential, dose duration, dosage quantity, and antioxidant safety, should be included in forthcoming clinical research.

## 7. Conclusions and Future Prospects 

Hydrogen controls gene expression and the phenotypes that ameliorate ailing situations. It has been concluded that H_2_ intervention can control the production of inflammatory cytokines, reduce or prevent both in vitro and in vivo cellular apoptotic damages, and scavenge free radicals, demonstrating the therapeutic benefit of H_2_. We conclude that H_2_ diffuses into cells and reduces mitochondrial free radicals via transporting electrons from damaged mitochondrial membranes. It also influences oxidative stress, hypoxia, and Nrf2 transcription. H_2_ has also been reportedly found to inhibit the nucleus transcription of anti-inflammatory NF-B and Foxp3. However, this could directly affect how Caspase3 and Bax are assembled, blocking apoptosis. This review has also found that molecular hydrogen therapies effectively remediated the life-threatening consequences of SARS-CoV-2 infection. In patients with mild-to-moderate disease symptoms, H_2_ administration has been reported to improve recovery through the abatement of the hyperinflammatory cytokine cascade and a reduction in inhalation resistance, as it functions as an effective anti-inflammatory and antioxidative agent. In essence, molecular hydrogen’s antioxidant capacity and respiratory disease studies suggest that inhaling it may also help in mitigating COVID-19. Despite its potency, molecular hydrogen needs to be further identified, characterized, described, and verified through pragmatic human and animal experiments, which may quadruple its significance as a novel and potential antioxidant agent.

Hence, this study details the current and plausible future prospective advancements in the field, based on the numerous therapeutic, nutraceutical, and pharmaceutical effects of H_2_ on the treatment and prevention of various diseases. Further studies are, however, required, since there are critical differences and clear disparities between clinical trials and efficacy testing conducted on animals and/or in cell cultures. Evidently, it has been found that not every study produced corroborating results. It is, however, recommended that clinical research should include data on the excess accumulation, reduction potential, dose duration, dosage quantity, and antioxidant safety of H_2_.

## Figures and Tables

**Figure 1 biomedicines-11-01892-f001:**
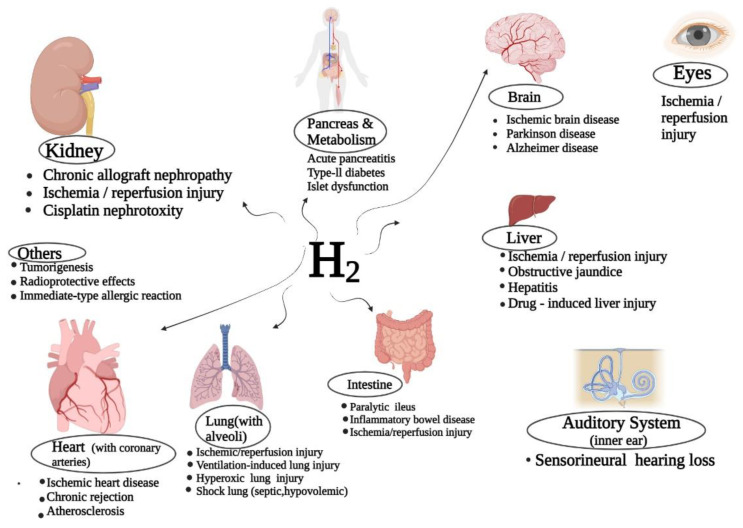
Therapeutic applications of hydrogen molecules (H_2_) in human beings.

**Figure 2 biomedicines-11-01892-f002:**
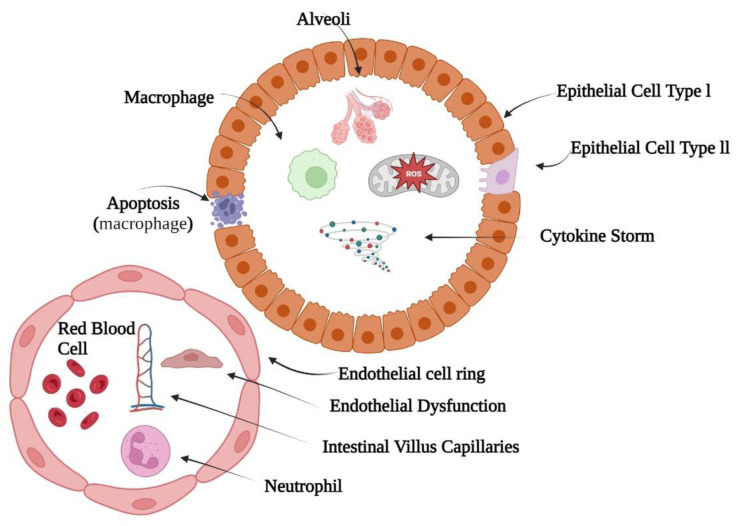
Dysregulated inflammation and malfunctioning of the alveolar and endothelial barriers involved in the pathophysiology of acute lung damage [24].

**Figure 3 biomedicines-11-01892-f003:**
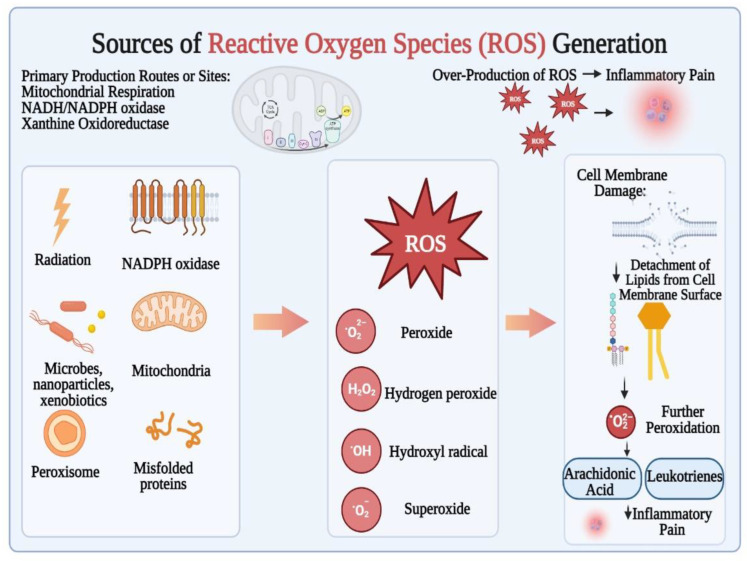
Production routes, sources, and effects of over-production of ROS (reactive oxygen species).

**Figure 4 biomedicines-11-01892-f004:**
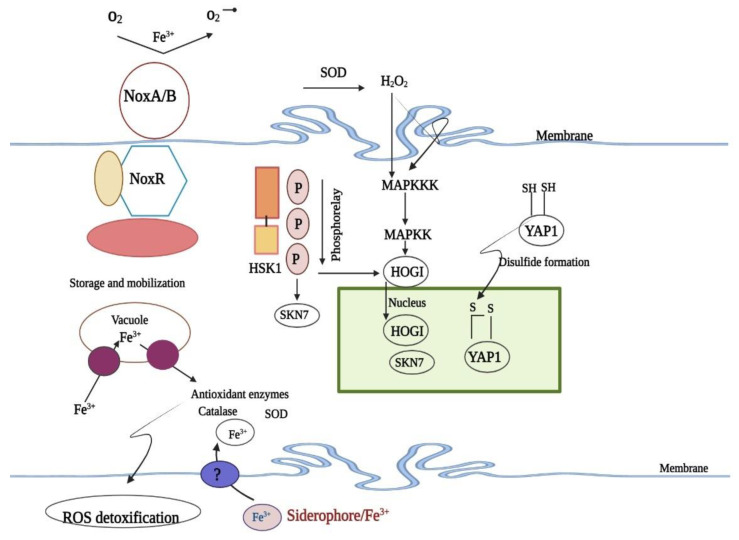
ROS Detoxification Pathway [48].

**Figure 5 biomedicines-11-01892-f005:**
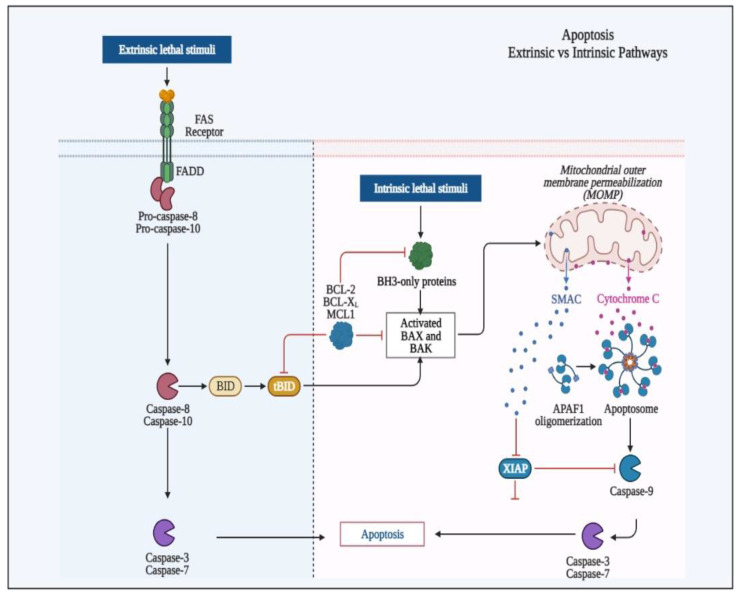
Induction of apoptosis through apoptotic pathways (i.e., intrinsic, and extrinsic) and subsequent activation of Caspase-3 and -7.

**Figure 6 biomedicines-11-01892-f006:**
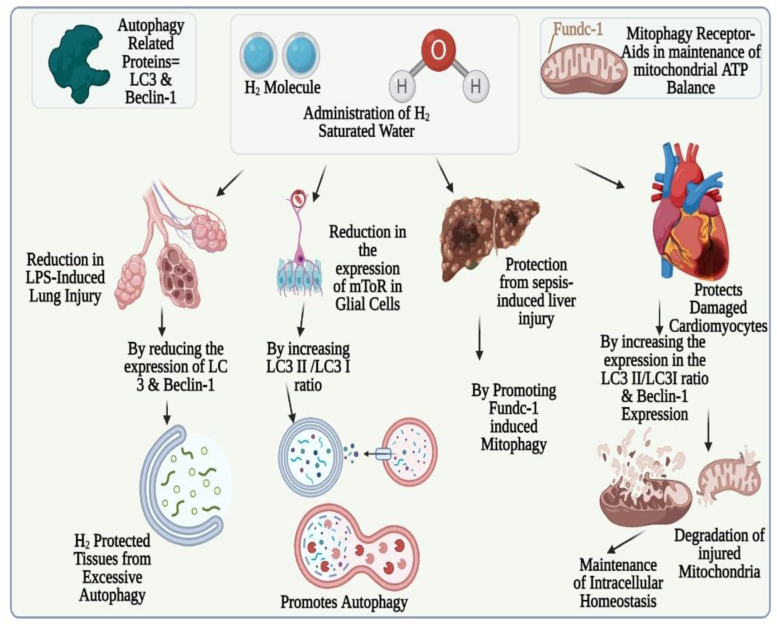
Bi-directional regulatory effect of hydrogen molecule on autophagy. Pictorial representation of inhibitory role of H_2_ in protecting the lung tissues from excessive autophagy; its role in promoting autophagy and mitophagy to protect the glial cells, injured liver cells, and damaged cardiomyocytes.

**Figure 7 biomedicines-11-01892-f007:**
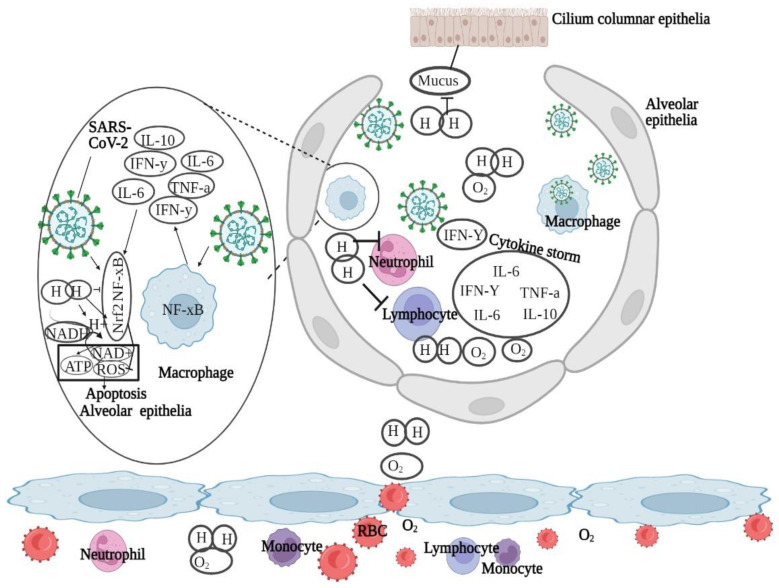
Hypothetical schematic illustration of hydrogen therapy for COVID-19 [94].

**Table 1 biomedicines-11-01892-t001:** Administration routes of molecular hydrogen in human bodies to combat various infections and/or diseases, along with the respective management strategies.

Administrative Routes in the Body	Subject/s (Time Taken to Initiate an Action)	Human Body Response/s	Effects on Target Organ or at Injury Site	Administration Protocol	Advantages	Prospective Risks Associated	Ref.
Dissolved H_2_ Saline	Rats (24 h)	Anti-inflammatory and anti-apoptotic effect	Myocardial I/R injury	10 mL/kg, 0.6 mmol/L,	Direct exposure or inoculation of dose at the target site	Cross-infection, Invasive	[30]
	Mice (12 h)	Anti-inflammatory response, reduces sepsis associated diseases	Encephalopathy	5 mL/kg, 0.6 mmol/L	Direct exposure or inoculation of dose at the target site	Cross-infection, Invasive	[31]
Drinking of dissolved H_2_ water	Human (2 weeks)	Alleviates Injuries	Injured soft tissues (sports-related)	2 g/day, H_2_-rich tablets	Safe and portable	Dose intake limitations	[32]
	Human (4 weeks)	Reduction in inflammation and anti-apoptotic	Peripheral blood vessels and blood cells	1500 mL/day, 0.753 mg/L	Safe and portable	Dose intake limitations	[33]
	Human (8 weeks)	Improves parapsoriasis	Plaques	10–15 min bathing with H_2_ water (two times a week)	Safe and portable	Dose intake limitations	[11]
	Guinea Pig (10 days)	Immunoregulation and improves allergic rhinitis	Allergic rhinitis	0.6 mmol/L, 20 μL/day Inoculated through nasal passage	Safe and portable	Dose intake limitations	[34]
	Mice (10 days)	Anti-inflammatory response	EAE ^1^ symptoms;	0.89 mM/0.36 Twice/day	Safe and portable	Dose intake limitations	[35]
H_2_ gas inhalation through nasal routes	Rats (120 min)	Antioxidant, protects from cerebral injury	Cerebral injury (I/R)	4, 1, or 2% H_2_	Dose and intake time can be ensured	If concentration rises above 4%, it may be explosive	[2]
	Rats (4 months)	Anti-inflammatory, ameliorates COPD	COPD ^2^ symptoms	2, 22 or 41.6% H_2_ For 2 h (Once/day)	Dose and intake time can be ensured	If concentration rises above 4%, it may be explosive	[36]
	Human (7 days)	Anti-inflammatory, ameliorates COPD	COPD symptoms	6 to 8 h/d, 66.6% H_2_	Dose and intake time can be ensured	If concentration rises above 4%, it may be explosive	[37]
	Human (daily till discharge)	Ameliorates COVID-19	COVID-19	66.6% H_2_ 33.3% O_2_	Dose and intake time can be ensured	If concentration rises above 4%, it may be explosive	[38]
H_2_ administration into the body via nanoparticles	Rats (3/24 h)	Antioxidant, anti-inflammatory, ameliorates lung and myocardial injuries	Myocardial injury (I/R), lung injury	4 × 109 or 2 × 1010 bubbles	Safe to use, high H_2_ content/unit volume	Expensive	[10]

^1^ EAE: experimental autoimmune encephalomyelitis; ^2^ COPD: chronic obstructive pulmonary disease.

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
