# Peer review of "Hydrogen Therapy and Its Future Prospects for Ameliorating COVID-19: Clinical Applications, Efficacy, and Modality"

_biomedicines, 2023, doi:10.3390/biomedicines11071892_

Round 1

Reviewer 1 Report

The review do not present new aspects regarding the scientific literature inherent to the subject matter. In particular there is a high similarity with a review already published in 2021 (Frontiers | Hydrogen, a Novel Therapeutic Molecule, Regulates Oxidative Stress, Inflammation, and Apoptosis (frontiersin.org))

Author Response

Reviewer 1

Comments and Suggestions for Authors

The review do not present new aspects regarding the scientific literature inherent to the subject matter. In particular there is a high similarity with a review already published in 2021 (Frontiers | Hydrogen, a Novel Therapeutic Molecule, Regulates Oxidative Stress, Inflammation, and Apoptosis (frontiersin.org))

Response: Dear reviewer, thank you so much for your valuable comments and feedback. We have revised the content of review article and added some more information. We believe that, the review covers a very good aspect for the possible use of Hydrogen as a therapeutic option for many diseases including infectious diseases.

Dear reviewer, before submitting the article to the journal, we have checked the similarity index using Turnitin and we found that the total similarity was 6%, however the requirement from many publishers is that it should be < 19%. The article you are talking about was showing the similarity of 3%, which we think that is fine for the review as it was below the maximum similarity limit by each of the article. However, to address your concern and in order to make it more suitable, we have revised the similarity and checked again using Turnitin.

In the last version, the percentage similarity between the two articles was merely 3% which has been further reduced now to 1% and the overall similarity is just 3%.

Reviewer 2 Report

I reviewed the manuscript titled “Application, Efficacy, and Modality of Novel Therapeutic Hydrogen Molecule in Oxidative Stress, Inflammation, and Apoptosis and Positive Future Prospects for Ameliorating COVID-19

Title is too clumsy. I suggest authors to modify

Line 23: H must be in lower-case

Authors ended up with review objectives. Authors should provide the results of the review and conclusions of the review

Introduction of the review is very short and failed to provide a sufficient background on performing this study

Authors must revise this carefully by providing background of the study; need of performing this review; examples related to the study and very clear review objectives

Figure 1: font size is extremely low. Authors must improve the quality of the manuscript

Figure 2. authors must provide the copyright statements, if needed

Figure 3: font size must be improved

3. Biological effects of hydrogen: Table should be provided by summarizing the biological effects of hydrogen. Authors described only two: antioxidant and anti-inflammatory. I suggest providing more details and other biological effects of H2.

Figure 5: font size must be improved

Figure 6: font size must be improved

Figure 7. authors must provide the copyright statements, if needed

Section 5 is well written; however, the latest literature in this area must be covered.

6. Effects on the Immune System: in-depth discussion must be provided

Conclusions must be revised and highlight the review findings and provide the recommendations/future prospects of the review.

References must be according to the journal format. 

Author Response

Reviewer 2

Comments and Suggestions for Authors

I reviewed the manuscript titled “Application, Efficacy, and Modality of Novel Therapeutic Hydrogen Molecule in Oxidative Stress, Inflammation, and Apoptosis and Positive Future Prospects for Ameliorating COVID-19

Title is too clumsy. I suggest authors to modify

Response: Dear reviewer, thank you so much for your valuable comments. Title has been modified as “Hydrogen therapy and its future prospects for ameliorating COVID-19: Clinical applications, efficacy, and modality”.

Line 23: H must be in lower-case.

Response:  The recommended changes have been updated in line 23 (current line number 24).

Authors ended up with review objectives. Authors should provide the results of the review and conclusions of the review

Response:  Conclusion and prospects have been updated as per the recommendations.  (New line numbers 576-601 in the revised manuscript).

Introduction of the review is very short and failed to provide a sufficient background on performing this study.

Response:   We appreciate your feedback. New paragraphs have been added in the introduction to aptly support and strengthen background knowledge. (New line numbers 77-121 in the revised manuscript).

Authors must revise this carefully by providing background of the study; need of performing this review; examples related to the study and very clear review objectives

Response: Thank you so much for your valuable feedback. We have updated the recommended changes in the manuscript which may help the readers to comprehend the background (line numbers 77-121), objective (118-125), and findings and future prospects of the study (576-600).

Figure 1: font size is extremely low. Authors must improve the quality of the manuscript

Response:  Font size has been increased as per your valuable recommendations.

Figure 2. authors must provide the copyright statements, if needed

Response:  We appreciate your valuable feedback. Although Figure 2 has been redrawn, yet its reference has also been given underneath. As it is redrawn and amended, it doesn’t need any ethical or copyright statements.

Figure 3: font size must be improved

Response: Previous picture has been replaced with the newer one having larger font sizes. 

  1. Biological effects of hydrogen: Table should be provided by summarizing the biological effects of hydrogen. Authors described only two: antioxidant and anti-inflammatory. I suggest providing more details and other biological effects of H2.

Response: Thank you so much for your valuable feedback. Section 2 deals with the biological effects of hydrogen molecule. It’s title, however, has been modified to properly indicate its content. Again, thanks to your kind notice. It encapsulates the role of H2 molecule in human physiology and its subsequent effect on various tissues and organs of human body. Table 2 describes the various administration routes of hydrogen molecule illustrating the subsequent human body responses, associated advantages and prospective risks which have been reportedly associated with the administration of molecular hydrogen.

Figure 5: font size must be improved

Response: Font size has been increased as per your valuable recommendations.

Figure 6: font size must be improved

Response: Font size has been increased as per your valuable recommendations.

Figure 7. authors must provide the copyright statements, if needed

Response:  We appreciate your valuable feedback. Although Figure 7 has been redrawn, yet its reference has also been given underneath. As it is redrawn and amended, it doesn’t need any ethical or copyright statements.

Section 5 is well written; however, the latest literature in this area must be covered.

Response: Thank you so much for your appreciation and recommendation. It has been updated with recently published literature. Now it constitutes from 473-498.

  1. Effects on the Immune System: in-depth discussion must be provided

Conclusions must be revised and highlight the review findings and provide the recommendations/future prospects of the review.

Response: Thank you so much for your valuable feedback. Certain modifications have been done in section 6. Now it constitutes from 552-to-571).  Conclusion has been aptly updated and revised (581-586) and Future prospects (line number 591-597) have also been updated as per your valuable recommendations.

References must be according to the journal format. 

Response: References has been rearranged using Endnote according to the journal guidelines.

Reviewer 3 Report

This manuscript is entitled "The Therapeutic Effects of Hydrogen Molecule on Disease Models and Human Diseases". The manuscript presents valuable information and attempts to present an in-depth analysis of the potential benefits of hydrogen therapy in various diseases, including COVID-19.

I believe the content of the manuscript is sound, and the overall design and presentation of the information is easy to follow. However, some areas need slight adjustments to improve readability, and provide a more streamlined perspective. Here are my suggestions:

Language and Grammar:

The language in the manuscript is mostly clear. However, some sentences are a bit complex and long-winded. Breaking these down into shorter, more concise sentences may enhance readability and ensure clear communication of the ideas presented.

Figure Presentation:

The figures provided are highly informative and greatly complement the text. However, please ensure that all the figures are of high resolution and adequately labeled for ease of understanding.

Clarity of Data and References:

In several instances, the authors cite multiple references for a single statement. I recommend refining this to present the most representative and relevant references for each claim to prevent overwhelming the readers and improve the flow of information.

Discussion on limitations:

While the article discusses the potential of hydrogen therapy extensively, there is limited mention of potential limitations or challenges associated with this therapy. Addressing this aspect would give the paper a more balanced view and would be beneficial for future research in this field.

Conclusion:

The conclusion does a good job of summarizing the article's contents. However, I suggest adding a section on future prospects, discussing the areas of research that are yet to be explored and the direction this field is moving in, considering the current evidence and advancements.

Formatting:

Please ensure consistent formatting throughout the manuscript, especially in the referencing style and figure captions.

In conclusion, this is a well-researched and structured manuscript that provides a comprehensive overview of hydrogen therapy and its potential in treating various diseases. With minor revisions, I believe this manuscript will be a valuable addition to the literature in this field.

Author Response

Reviewer 3

Comments and Suggestions for Authors

This manuscript is entitled "The Therapeutic Effects of Hydrogen Molecule on Disease Models and Human Diseases". The manuscript presents valuable information and attempts to present an in-depth analysis of the potential benefits of hydrogen therapy in various diseases, including COVID-19. I believe the content of the manuscript is sound, and the overall design and presentation of the information is easy to follow. However, some areas need slight adjustments to improve readability, and provide a more streamlined perspective. Here are my suggestions:

Response: Dear Reviewer! We appreciate your kind words. We have worked hard to follow your kind suggestions to improve the quality of literature in the best interest of the readers.

Language and Grammar:

The language in the manuscript is mostly clear. However, some sentences are a bit complex and long-winded. Breaking these down into shorter, more concise sentences may enhance readability and ensure clear communication of the ideas presented.

Response: We appreciate your insightful remarks very much. Some of the longer, more complex lines have been revised and divided into shorter ones. This would benefit the readers and ensure that the basic ideas have been clearly communicated. (Line number 137-141, 230-233).

Figure Presentation:

The figures provided are highly informative and greatly complement the text. However, please ensure that all the figures are of high resolution and adequately labelled for ease of understanding.

Response: Dear Reviewer! Thank you for your complement. It means a lot to us.  However, we have rechecked and ensured that every figure is adequately labelled and maintains highest possible resolution.

Clarity of Data and References:

In several instances, the authors cite multiple references for a single statement. I recommend refining this to present the most representative and relevant references for each claim to prevent overwhelming the readers and improve the flow of information.

Response: Dear Reviewer! Thank you so much for your feedback. More than one references for a single statement entails that the same concept has been cited in multiple research publications. However, to prevent any future confusion for the readers such references have been converted to the most relevant single reference.  (Line number 170-174, 285-293,309-313, 489-492, 550-554).

Discussion on limitations:

While the article discusses the potential of hydrogen therapy extensively, there is limited mention of potential limitations or challenges associated with this therapy. Addressing this aspect would give the paper a more balanced view and would be beneficial for future research in this field.

Response: The potential application, future prospects, and challenges of hydrogen therapy has been discussed in the manuscript. (Line Number: 277-279,302-306,355-356,388-390,431-435).

Conclusion:

The conclusion does a good job of summarizing the article's contents. However, I suggest adding a section on future prospects, discussing the areas of research that are yet to be explored and the direction this field is moving in, considering the current evidence and advancements.

Response: Respected Reviewer! Thank you so much for your valuable suggestion. We have updated a paragraph indicating future perspective and challenges ahead.  (Lines number 596-603).

Formatting:

Please ensure consistent formatting throughout the manuscript, especially in the referencing style and figure captions.

Response: Thank you so much for your positive feedback. We have formatted the whole manuscript in single referencing style and figure captions.

In conclusion, this is a well-researched and structured manuscript that provides a comprehensive overview of hydrogen therapy and its potential in treating various diseases. With minor revisions, I believe this manuscript will be a valuable addition to the literature in this field.

Response: Dear Reviewer! We appreciate your kind words and positive feedback, which gave us encouragement. To the best of our capacity, we have tried to implement your helpful suggestions. We anticipate that this manuscript will be very beneficial to the in comprehending the applicability and efficacy of hydrogen molecules in therapeutics and nutraceuticals. Additionally, it would assist community and health experts in identifying plausible solutions to the challenges ahead in future.

Round 2

Reviewer 1 Report

The overall recommendation for the review is rejected. Although the text does not exceed the limits of similarity accordingly to authors,  the scientific contents of the review do not present a sufficient level of originality and do not offer a significant contribution to the field.

Author Response

Reviewer 1

Comments and Suggestions for Authors

The overall recommendation for the review is rejected. Although the text does not exceed the limits of similarity accordingly to authors, the scientific contents of the review do not present a sufficient level of originality and do not offer a significant contribution to the field.

Response: Dear reviewer, thank you so much for your valuable comments and feedback. We have revised the content of review article and added some more information. We believe that, the review covers a very good aspect for the possible use of Hydrogen as a therapeutic option for many diseases including infectious diseases. To the best of our capacity, we have tried to implement your helpful suggestions. We anticipate that this manuscript will be very beneficial to the in comprehending the applicability and efficacy of hydrogen molecules in therapeutics and nutraceuticals. Additionally, it would assist community and health experts in identifying plausible solutions to the challenges ahead in future. We have highlighted the new changes in yellow colour for your reference.

The following changes has been made in the revised version of manuscript:

  1. This would benefit the readers and ensure that the basic ideas have been clearly communicated. (Line number 137-141, 230-233).
  2. The potential application, future prospects, and challenges of hydrogen therapy has been discussed in the manuscript. (Line Number: 277-279,302-306,355-356,388-390,431-435).
  3. We have updated a paragraph indicating future perspective and challenges ahead. (Lines number 596-603).